# Immunomodulatory Factor TIM3 of Cytolytic Active Genes Affected the Survival and Prognosis of Lung Adenocarcinoma Patients by Multi-Omics Analysis

**DOI:** 10.3390/biomedicines10092248

**Published:** 2022-09-10

**Authors:** Liusheng Wu, Yanfeng Zhong, Dingwang Wu, Pengcheng Xu, Xin Ruan, Jun Yan, Jixian Liu, Xiaoqiang Li

**Affiliations:** 1Peking University Shenzhen Hospital, Clinical College of Anhui Medical University, Tsinghua university, Shenzhen 518036, China; 2Thoracic Surgery, Peking University Shenzhen Hospital, Shenzhen 518036, China

**Keywords:** cytolytic activity, lung adenocarcinoma, immunoregulatory factor, immune checkpoint, multi-omics study, drug sensitivity

## Abstract

[**Objective**] Using multi-omics research methods to explore cytolytic activity-related genes through the immunoregulatory factors HAVCR2 (TIM3) affecting the survival and prognosis of lung adenocarcinoma. [**Methods**] We combined Cox single factor regression and lasso regression feature selection algorithm to screen out the key genes of cytolytic activity in lung adenocarcinoma, and applied multi-omics research to explore the clinical predictive value of the model, including onset risk, independent prognosis, clinical relevance, signal transduction pathways, drug sensitivity, and the correlation of immune regulatory factors, etc. TCGA data are used as the experimental group, and GEO data is used as the external data control group to verify the stability of the model. The survival curve was generated by the Kaplan–Meier method and compared by log-rank, and the Cox proportional hazard model was used for multivariate analysis. In this study, 10 fresh tissue samples of lung adenocarcinoma were collected for cellular immunohistochemical experiments to analyze the expression of immunoregulatory factors in cancer tissues, and the key immunoregulatory factors were verified and screened out. [**Results**] A total of 450 genes related to cytolytic activity were differentially expressed, of which 273 genes were up-regulated and 177 genes were down-regulated. A total of 91 key genes related to cytolytic activity related to the prognosis of lung adenocarcinoma were screened through Cox single factor regression. The ROC curve results showed that the AUC values of 1, 3, and 5 years in the training set and test set were all greater than 0.7, indicating that the model has a valid verification. The level of risk score is significantly related to the sensitivity of patients to AKT inhibitor VIII, Lenalidomide, and Tipifarnib. In addition, our study also found that receptor and MHC genes related to immunomodulatory, and chemokines, including HAVCR2, are more highly expressed in the low-risk group. [**Conclusions**] HAVCR2 (TIM3) immunoregulatory factors affect the expression of key genes that affect cytolytic activity in lung adenocarcinoma cells, and to some extent indirectly affect the survival and prognosis of patients with lung adenocarcinoma.

## 1. Introduction

Lung cancer is one of the malignant tumors with the fastest increases in morbidity and mortality and represents the biggest threat to people’s health and life. Lung cancer has changed from a rare disease to a global and public health problem [1]. The causes of lung cancer have become more complex with the world’s industrialization, urbanization, and environmental pollution. In the last 50 years, many countries have reported a significant increase in the incidence and mortality of lung cancer, but the cause of lung cancer is still not completely clear [2,3,4]. Therefore, screening biomarkers to predict the early progression of lung cancer and establishing a risk prediction model for lung cancer are crucial for disease management. In addition, immunotherapy is being explored as an alternative to adjuvant therapy due to poor prognosis after standard treatment. Immune checkpoint is a class of immunosuppressive molecules, which can regulate the intensity and extent of the immune response, so as to avoid the damage and destruction of normal tissues. During the occurrence and development of tumors, the immune checkpoint is one of the main reasons for immune tolerance [5]. Today’s checkpoint inhibitor drugs target receptors, such as PD-1 and cytotoxic T lymphocyte-associated antigen-4 (CTLA-4), which act as a kind of “off switch” on a T cell’s surface to prevent it from attacking other cells. The use of one or more of these drugs can prevent the pathways from releasing the inhibitors so that the immune system can fight the tumor. However, despite the great success of anti PD-1/PD-L1 immune checkpoint antibodies in cancer treatment over the years, more than half of patients who rely on these drugs have relapsed or their cancer has progressed. Studies [6,7] have shown that cytotoxic T cells, natural killer cells, and other immune cells reflect the strength of anti-tumor immunity and are associated with the effectiveness of immune checkpoint inhibitors. In addition, they are closely associated with the clinical prognosis of many tumors, including lung and colorectal cancer. Cytolytic activity, as a new biomarker for immunotherapy, can characterize the anti-tumor immunoactivity of immune cells, such as CD8^+^ cells, cytotoxic T cells, and macrophages [8]. Therefore, it is necessary to explore genes and predictive models related to the CYT level to provide a basis for the prognosis management of lung cancer.

Immune checkpoint receptor protein TIM3 is a type I membrane protein, also known as hepatitis A virus cell receptor 2 (HAVCR2), which is a negative regulator of anti-tumor immunity. As the largest database of cancer gene information at present, the comprehensive TCGA is not only reflected in many cancer types but also reflected in multi-omics data, including gene expression data, miRNA expression data, copy number variation, DNA methylation, and SNPs [9]. In the early stages, we screened genes related to cytolytic activity [10], constructed a model related to cytolytic activity through TCGA dataset, verified the stability of the model in three GEO datasets, and predicted the prognosis of lung cancer patients through the model related to cytolytic activity, which will provide a new basis for the management of the disease.

## 2. Materials

### 2.1. Data Acquisition

The TCGA database (https://portal.gdc.cancer.gov/) (accessed on 1 January 2022). is the biggest cancer gene information database, including gene expression data, the miRNA expression data and copy number variation, DNA methylation, SNPS, and other data. We downloaded the original mRNA expression data of processed LUAD, including the normal group (*n* = 59) and the tumor group (*n* = 535). The Limma package was used to integrate and standardize the FPKM data of level 3 mRNA, which was downloaded to analyze the differently expressed genes and their expression levels [11]. The screening conditions of different genes were LogFc > 1 and *p* < 0.05. The Series Matrix File data File of GSE37745 from the NCBI GEO public database was downloaded, the platform GPL570 was annotated, and the data of 106 LUAD patients with the complete expression profile and survival information was downloaded. The data of 83 LUAD patients with complete expression profile and survival information were extracted [12]. The Series Matrix File data File of GSE50081 was downloaded, the platform GPL570 was annotated, and the data of 127 LUAD patients with complete expression profile and survival information was extracted. GeneCards (https://www.genecards.org/) (accessed on 1 January 2022) were used to obtain 976 cytolysis activity-related gene sets through our database.

### 2.2. Go and KEGG Functions

Clusterprofiler (R 3.6.3, The United States) was used to functionally annotate the difference factors to comprehensively explore the functional correlation of these prognostic genes [13]. Gene Ontology (GO) and the Kyoto Encyclopedia of Genes and Genomes (KEGG) were used to assess related functional categories. The GO and KEGG enrichment pathways with *p* and Q values of less than 0.05 were considered as significant categories.

### 2.3. Model Construction and Prognosis

Differential genes related to cell lytic activity were selected and lasso regression was used to further construct prognostic correlation models. After the expression values for each specific gene were included, a risk score formula for each patient was constructed and weighted with its estimated regression coefficients in lasso regression analysis [14,15]. According to the risk scoring formula, patients were divided into low risk group and high risk group with the median risk score as the cut-off point. Survival differences between the two groups were assessed by Kaplan–Meier and compared using log-rank statistical methods. Lasso regression analysis and stratified analysis were used to examine the role of risk score in predicting patient outcomes, and the ROC curve was used to study the accuracy of model prediction.

### 2.4. Drug Sensitivity Analysis

Using genomics based on the largest drug database (GDSC—Genomics of Drug Sensitivity in Cancer, https://www.cancerrxgene.org/) (accessed on 1 January 2022), we utilized the R software(R 3.6.3, The United States) package “pRRophetic” to predict chemotherapy sensitivity of each tumor samples; the estimates of IC50 for each specific chemotherapeutic agent were obtained by regression method [16], and 10 cross-validations were performed with the GDSC training set to test the regression and prediction accuracy [17]. Default values were selected for all parameters, including “combat”, which removes the batching effect, and the average of the duplicate gene expression.

### 2.5. Analysis of Immune Cell Infiltration

The Cibersort algorithm was used to analyze the RNA-Seq data of LUAD patients in different subgroups to infer the relative proportion of 22 kinds of immunoinfiltrating cells [18], and Spearman correlation analysis was performed on the gene expression level and the content of immune cells, and *p* < 0.05 was considered statistically significant.

### 2.6. GSEA Enrichment Analysis

The expression profiles of lung cancer patients were analyzed by gene collection enrichment (GSEA, http://www.broadinstitute.org/gsea) (accessed on 1 January 2022) to determine the high risk and low risk of differentially expressed genes between groups of patients [19]. The gene set was filtered using a maximum and minimum gene set size of 500 and 15 genes, respectively. After 1000 permutations, a rich gene set was obtained based on *p* < 0.05 and a false discovery rate (FDR) = 0.25.

### 2.7. Statistical Analysis

Survival curves were generated through the Kaplan–Meier method and compared using log-rank analysis. The Cox proportional risk model was used for multivariate analysis. All statistical analyses were conducted in the R software(R 3.6.3, The United States). All statistical tests were bilateral, and *p* < 0.05 was statistically significant.

## 3. Results

### 3.1. To Investigate the Expression Profile and Related Signaling Pathways of Genes Related to Cytolytic Activity in LUAD Cohorts

We downloaded raw LUAD mRNA expression data (FPKM) processed from the TCGA database and extracted 935 sets of regulatory factors related to cytolytic activity. Differential expression analysis was performed between lung cancer patients and non-lung cancer patients using the Limma package [20]. The results showed that 450 genes related to cell lytic activity were differentially expressed (logFc > 1 and logFc < −1 and *p* < 0.05), 273 of which were upregulated. There were 177 down-regulated genes (Figure 1a), and a protein interaction network analysis of genes in different gene concentrations was conducted by Cytoscape software (R 3.9.0, The United States) (Figure 1b). GO and KEGG enrichment analysis showed that these differential genes were significantly enriched in a large number of pathways. For example, there is the positive regulation of cell activation, the external side of plasma membrane, peptide antigen binding, and the regulation of GO-rich concentrations. A large number of genes were enriched in lymphocyte activation and other pathways (Figure 1c). In the process of KEGG enrichment, several pathways, primary immunodeficiencies, and Leukocyte transmigrations were found. At the same time, a large number of genes were enriched in metabolism-related pathways (Figure 1d).

### 3.2. Obtain Prognostic Related Genes and Build a Prediction Model

To further identify the key genes in the cytolytic activity gene concentration, we collected clinical information from LUAD patients, and we used a combination of Cox univariate regression and lasso regression feature selection algorithm to screen out the characteristic genes in lung cancer (Figure 2a,b). The results showed that a total of 91 prognostic related genes were screened out by Cox univariate regression (Figure 2c). We randomly divided TCGA patients into a training set and a validation set at a 4:1 ratio. After lasso regression analysis, the optimal risk score value corresponding to each sample was obtained for subsequent analysis. According to the median risk score, LUAD patients were divided into high-risk and low-risk groups and analyzed using Kaplan–Meier curves [21]. The OS of the high-risk group in both the training set and the test set was significantly lower than that of the low-risk group (Figure 2d,e). In addition, the results of ROC curve showed that the AUC values of 1, 3, and 5 years in the training set and the test set were all greater than 0.7 (Figure 3a,b), indicating that the model had a good verification efficiency.

### 3.3. The Clinical Predictive Value of the Model Was Discussed in Multi-Omics Study

The tumor microenvironment is mainly composed of tumor-related fibroblasts, immune cells, extracellular matrix, a variety of growth factors, inflammatory factors, special physical and chemical characteristics, and the cancer cells themselves. The tumor microenvironment significantly affects the diagnosis, survival outcome, and clinical treatment sensitivity of tumors [22,23,24]. By analyzing the relationship between the risk score and tumor immune invasion, the potential molecular mechanism of the risk score affecting the progression of lung cancer was further explored (Figure 3c). Early lung cancer surgery combined with chemotherapy has a clear effect. Based on the drug sensitivity data from the GDSC database [25], our study predicted the chemotherapy sensitivity of each tumor sample through the R package “pRRophetic” and further explored the risk score and the sensitivity of common chemotherapy drugs. Results showed that risk score was significantly correlated with patients’ sensitivity to AKT VIII, Lenalidomide, and Tipifarnib (Figure 3d). The results showed that risk score was significantly positively correlated with macrophages M0, CD4 memory-activated T cells, macrophages M1, neutrophils, memory-activated NK cells and B cells, monocytes, resting mast cells [26], resting dendritic cells (Figure 3e), and the regulation network among immune cells is shown in Figure 3f. We further explored the mutation profiles of patients in the low-risk group, and the results showed that the proportion of mutations in TP53 and other genes in the high-risk group was significantly higher than that in the low-risk group (Figure 3g). Meanwhile, we also found that the tumor mutation load in the high-risk group was significantly higher than that in the low-risk group (Figure 4a).

### 3.4. Discussion on Specific Signaling Mechanisms Related to the Prognostic Model

We then investigate the specific signaling pathways involved in the high-risk correlation model and explore the potential molecular mechanisms [27] by which the risk score influences tumor progression. The GO results of GSEA analysis show that the differences between the two groups were mainly enriched by mast cell-mediated immunity, mitotic nuclear division, vacuolar acidification and other signaling pathways. The KEGG results of GSEA analysis showed that the cell cycle, intestinal immune network for IgA production, oocyte meiosis, the P53 signaling pathway and primary bile were mainly enriched in different pathways between the two groups’ acid biosynthesis, RNA degradation, and other signaling pathways (Figure 4a,b).

### 3.5. External Datasets Were Used to Verify the Robustness of the Prognostic Model

We downloaded the data of LUAD patients with survival data processed from the GEO database (GSE37745, GSE30219 and GSE50081), predicted the clinical classification of LUAD patients in the GEO database according to the model, and evaluated the survival difference between the two groups using Kaplan–Meier [28,29,30]. The stability of the prediction model is discussed, and results showed that the OS in the high-risk group of the two GEO external validation sets was significantly lower than that in the low-risk group (Figure 4c–e). We used external datasets to conduct ROC curve analysis on the model, and the results showed that the model had a strong predictive effect on the prognosis of patients (AUC values of 1, 3, and 5 years in the GSE37745 dataset were all greater than 0.6; AUC values of 1, 3, and 5 years in the GSE30219 dataset were all greater than 0.6; AUC values of 1, 3, and 5 years in the GSE50081 dataset were all greater than 0.6) (Figure 4f–h).

### 3.6. Analysis of Risk and Independent Prognosis

The results of logistic regression analysis showed that in all our samples, the risk score value of lung cancer made a significant contribution to the scoring process of predictive analysis [31]. Calibration curves for the nomogram prediction model are also drawn to indicate the stability of the nomogram prediction model (Figure 5a,b). In addition, univariate and bivariate analyses showed that the risk score was an independent prognostic factor in LUAD patients (Figure 6a,b).

### 3.7. Correlation Analysis between Risk of Disease and Multiple Clinical Indicators

According to the size of clinical index values, the corresponding risk score values of the samples were divided into different groups, and the grouping results of each clinical index were shown in the form of a boxplot (Figure 6c–h). Rank sum test (Kruskal test) found that the distribution of risk score values in gender, stage, T, M, N, and other clinical indicators were significant between groups (*p* < 0.05). The risk score has good applicability to the grouping of this group of samples.

### 3.8. Relationship between Risk Score Subtypes and Immunoregulatory Factors

The TIDE immunotherapy analysis was further performed in the high-risk groups, and the results showed significant differences in immune dysfunction and rejection between the high-risk groups (Figure 7a,b). The genes associated with immune modification include BTLA, CSF1R, ADORA2A, IL-10, LGALS9, HAVCR2, CD160, CTLA4, CD244, TGFB1, TGFBR1, and CD96 (Figure 7c). Tumor immunotherapy is a hotspot in current and future research, and T cell immunoglobulin mucin 3 (TIM3) is another emerging immune checkpoint molecule after PD-1/PD-L1 and CTLA-4 [32]. TIM3 expression on CD8^+^ T cells in the tumor microenvironment is considered to be a major marker of T cell dysfunction. Therefore, TIM3 is of priority for further study among the genes related to immune modification in this study.

### 3.9. High Expression of TIM3 in Lung Adenocarcinoma May Be Involved in Tumor Immune Response

In this study, the high and low expression of TIM3 in lung adenocarcinoma tissues and adjacent tissues was verified by PCR experiments (Figure 8). The results of real-time fluorescence quantitative PCR were generally presented in the form of mean ± standard deviation. The results are usually analyzed by the method of relative definite halo analysis, the expression level of the target gene is obtained by 2^–ΔΔCTmethod^, β-actin is the internal reference gene, and TIM3 is the target gene. The expression level of each gene is measured twice, so the final CT value is taken as the average of the results of the two groups. For the calculation of the ΔΔ Ct value, the results of the final samples and the normal control group 2^–ΔΔCT^ are the expressions of the multiple differences between the two groups. In this study, relative gene expression analysis was generally performed using the 2^–ΔΔCT^ method, which was simple and easy to undertake. The necessary conditions for this analysis were when the amplification efficiency of the target gene and reference gene was close to 100% and the efficiency deviation between them was within 5%.

The final QRT-PCR results indicated that there was a statistically significant difference in the related mRNA levels of TIM3 in lung adenocarcinoma tissues and normal tissues (*p* < 0.001).

### 3.10. Immunohistochemistry and Immunofluorescence Assay

TIM3 showed two different states of high and low expression, and the positive signal was mainly located in the nucleus, which was consistent with its function as a gene regulatory factor, and also reflected the relationship between the TIM3 expression level and the clinicopathological characteristics of lung adenocarcinoma patients (Figure 9). Immune checkpoint receptor protein TIM3 is a type I membrane protein, also known as hepatitis virus cell receptor 2 (HAVCR2), which is a negative regulator of anti-tumor immunity. TIM3 is a member of the TIM family, which is composed of eight TIM1-TIM8 members. TIM3 is a type I membrane glycoprotein expressed in terminally differentiated CD4^+^T cell subsets (e.g., Th1 cells, Th17 cells, and Tregs) and CD8^+^T cell subsets type 1 CD8^+^T cells (Tc1) but not in Th2 cells. It is also expressed in B cells, macrophages, dendritic cells, natural killer cells, mast cells, and monocytes (Figure 10).

## 4. Discussion

Today, tumors have become the main cause of death worldwide. However, whether in terms of morbidity or mortality, lung adenocarcinoma has shown an increasing trend year by year. Although the level of medical care has greatly improved in recent years [33], which has played an important role in the early screening, diagnosis, and treatment of lung cancer, the degree of malignancy of lung cancer is relatively high and the survival rate is still very low. Therefore, research on the pathogenesis of lung adenocarcinoma and the management of prognosis assessment must be strengthened, and the importance of establishing a risk model for lung cancer management must be further clarified so as to achieve early prevention and effective treatment [34].

In recent years, immunotherapy for the programmed death factor 1 (PD-1) and its ligand 1 (PD-L1) has made rapid progress in lung adenocarcinoma [35]. Although immunotherapy has a significant effect, only a small number of patients can benefit from it. How to choose effective biomarkers to screen out potential benefit groups is the main problem currently facing. Some people think that its monitoring effect on tumors even outperforms T cells. As far as the current experiments and people’s understanding are concerned, the killing of tumor cells by macrophages is multi-method and multi-mechanism [36]. It is still difficult to use one mechanism to explain all phenomena, including cytolysis, autophagy, apoptosis, and other methods. When the cells of the body become cancerous, this induces immune cells to attack the tumor cells in a variety of ways, destroying the cell membrane through complement or cytolysin, etc., and causing the tumor cells to undergo a lytic effect, thereby achieving the ultimate goal of killing the tumor cells. However, this immune effect is different in different individuals or against different tumor cells, and its response efficiency is different in high and low expression. This is related to the mutated genes in tumor cells. Some mutated genes are closely related to cytolytic activity; relevant, highly sensitive, and specific. At present, many studies have begun to explore the expression of key genes related to cytolytic activity in different types of tumors, including cervical cancer, bladder cancer, prostate cancer, breast cancer, and lung adenocarcinoma [37]. Cytolysis is an effect mechanism for the body to kill viruses and other microorganisms infecting cells and tumor cells. Compared with the N-terminus, the cytolytic activity regulator contains a conserved C2 domain at the C-terminus, which is in the synaptic binding protein (SYNC2A), protein kinase CB (PKC-B), and phospholipase CD (PLC-D). There are also reports in China [38] that among these proteins, C2 interacts with the cell membrane to regulate critical intracellular reactions, including membrane transport, second messenger production, GTP ases activation, Ca2^+^-dependent neurotransmitter release, and control protein phosphorylation. It can be seen that the high expression of cytolytic activity genes plays a key role in tumor immunotherapy.

In this study, Cox single factor regression and lasso regression feature selection algorithms were used to screen out the key genes of cytolytic activity related to the prognosis of lung adenocarcinoma and construct a risk model. The median risk score in the test set was −0.0354256887557756, and patients were classified as a high-risk group and low-risk group, and Kaplan–Meier curve analysis was used. The OS of the high-risk group in the training set and the test set was significantly lower than that of the low-risk group. In addition, the ROC curve results show that the AUC values of 1, 3, and 5 years in the training set and the test set are all greater than 0.7, indicating that the model has good verification performance. The protein of HAVCR2 encoded by this gene belongs to the immunoglobulin superfamily [31,32] and TIM family of proteins. The CD4-positive T helper lymphocytes can be divided into types 1 (Th1) and 2 (Th2) on the basis of their cytokine secretion patterns [39]. This protein is a Th1-specific cell surface protein that regulates macrophage activation and inhibits Th1-mediated auto- and alloimmune responses and promotes immunological tolerance. The CD160 molecule is a glycosylphosphatidylinositol (GPI)-anchored protein, also known as BY55, which was first discovered on the surface of NK cells [33,34]. The CD160 gene is located in the 1q21.1 region of the human chromosome, and its protein molecular weight is a 27 kda glycoprotein. Its open reading frame is composed of 181 amino acids and it has weak homology with the kill-inhibitory receptor KIR on the surface of NK cells. Classical and non-classical MHC I molecules have low affinity with CD160, and this combination can enhance NK cells and the cytotoxic activity of T cells.

We further explored the risk score and the sensitivity of common chemotherapy drugs. The results of the study showed that the level of risk score was significantly related to the sensitivity of patients to AKT inhibitor VIII, Lenalidomide, and Tipifarnib. The results of the study showed that the risk score was significantly positively correlated with macrophages M0, memory-activated CD4 T cells, macrophages M1, neutrophils, and activated NK cells and significantly negatively correlated with memory B cells, monocytes, resting mast cells, and resting dendritic cells [40]. Through GO and KEGG enrichment analysis, we found that these differential genes are significantly enriched in a large number of pathways, such as the positive regulation of cell activation, the external side of the plasma membrane, peptide antigen binding, the regulation of lymphocyte activation, and other pathways in GO enrichment.

At present, target drugs represented by PD-1/PD-L1 have been gradually developed to improve the prognosis and prolong the survival time of patients with advanced cancer, but only a small proportion of patients show a long-term and lasting response. In addition, some patients may develop adaptive resistance to current immunotherapy regimens. As a promising inhibitory receptor in many emerging immune checkpoints, TIM3 has shown initial results in clinical trials as a monotherapy or in combination with anti-PD-1/PD-L1 drugs. The drug types have been expanded from a monoclonal antibody to dual antibody, and by all indications cover both hematological tumors and solid tumors (including lung cancer), but further studies are needed to determine the future direction of TIM3.

## 5. Conclusions

Immunoregulatory factor TIM3 may regulate the proliferation and invasion of lung adenocarcinoma cells through signaling pathways (such as cytolytic activity), further regulate the immune microenvironment of lung adenocarcinoma, and has a certain risk correlation with drug resistance of tumor cells, which indirectly affects the survival and prognosis of lung adenocarcinoma patients to a certain extent.

## Figures and Tables

**Figure 1 biomedicines-10-02248-f001:**
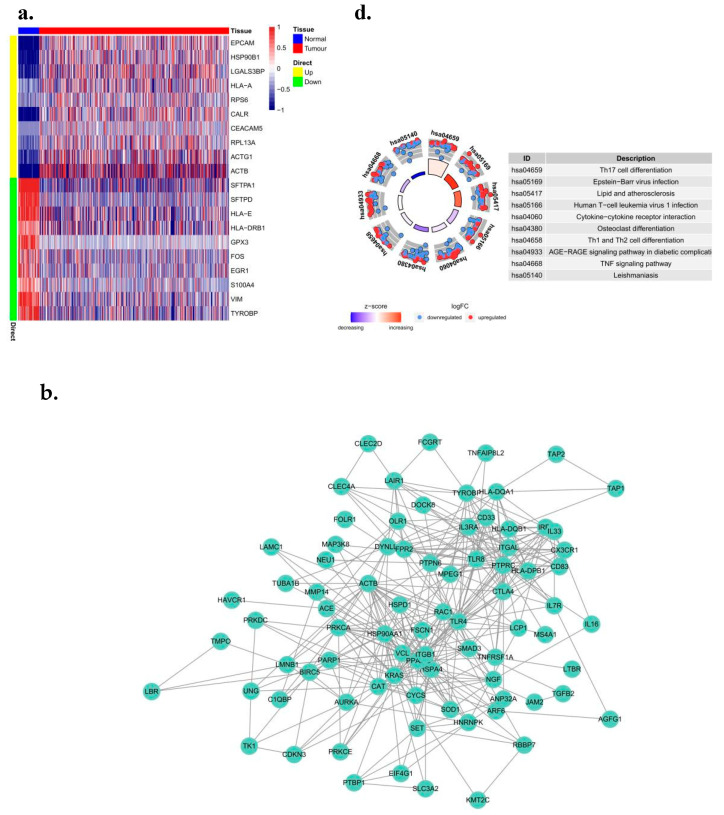
Identification and pathway enrichment analysis of differential genes associated with cytolytic activity in LUAD. (**a**) Differential genetic heat map. Heat maps of 10 up-regulated genes. (**b**) PPI analysis. Cytoscape software was used to analyze the protein interaction network of genes in different gene sets. (**c**) GO and (**d**) KEGG enrichment analysis.

**Figure 2 biomedicines-10-02248-f002:**
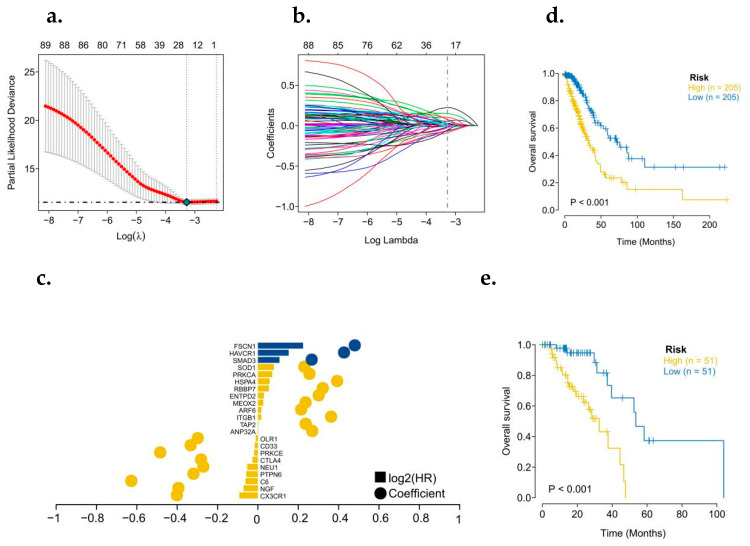
Construction of a prognostic model for genes associated with LUAD’s cytolytic activity. (**a**,**b**) Lasso regression was used to construct a prognostic Cytolytic activity-related prognostic model. (**c**) Model gene coefficient map. (**d**) Survival analysis of TCGA training set model. (**e**) Survival analysis of TCGA test set model (Yellow: High expression; Blue: Low expression).

**Figure 3 biomedicines-10-02248-f003:**
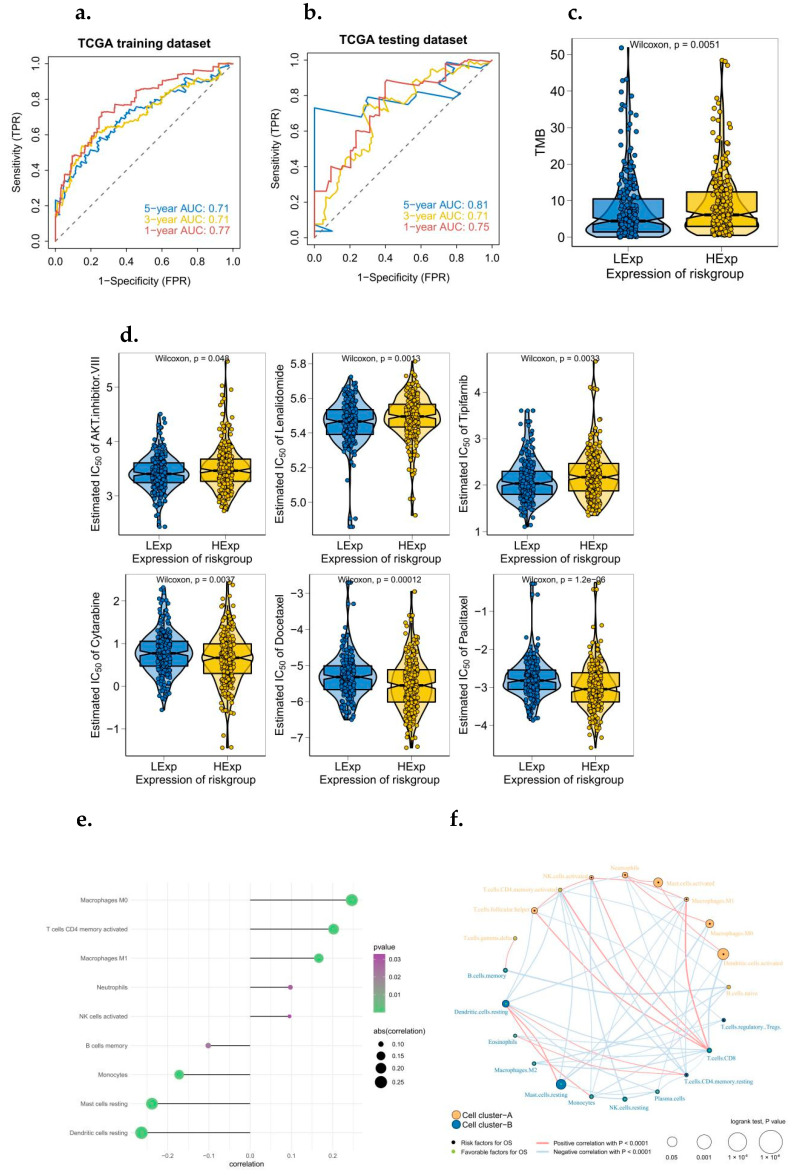
ROC analysis of a prognostic model associated with cytolytic activity, and the clinical predictive value of the model was investigated in multi-omics study. (**a**) ROC curve related to TCGA training set model (1, 3, and 5 years). (**b**) ROC curve related to TCGA test set model (1, 3, and 5 years). (**c**) Tumor mutation load was significant between high and low risk groups. (**d**) Risk score and sensitivity to common chemotherapeutic agents. (**e**) Relationship between risk score and tumor immunoinvasion. (**f**) Regulatory networks between immune cells. (**g**) Mutation profiles of patients at high or low risk.

**Figure 4 biomedicines-10-02248-f004:**
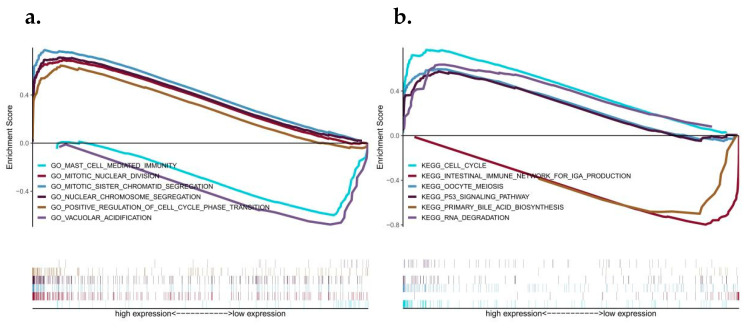
Analysis of GSEA pathways between high and low risk groups and external datasets verify the predictive effectiveness of the risk model. (**a**) GSEA–GO enrichment analysis related to high- and low-risk groups; (**b**) Correlation GSEA–KEGG enrichment analysis for high- and low-risk groups. (**c**–**e**) Kaplan–Meier curves of three GEO external datasets. (**f**–**h**) 1, 3, and 5 year ROC curves of three GEO external datasets.

**Figure 5 biomedicines-10-02248-f005:**
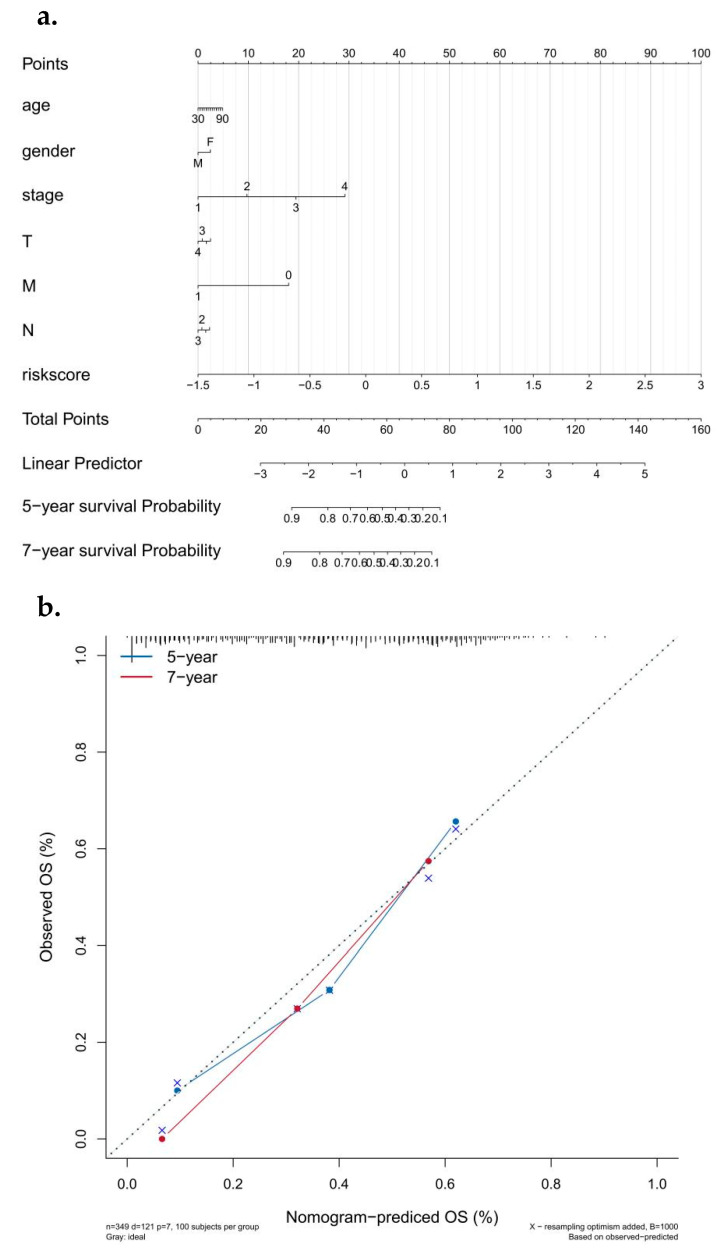
Construction and efficacy assessment of a nomogram model for risk scoring. (**a**) Column diagram related to the model. (**b**) Corrected curve of model related histogram (5 years/7 years).

**Figure 6 biomedicines-10-02248-f006:**
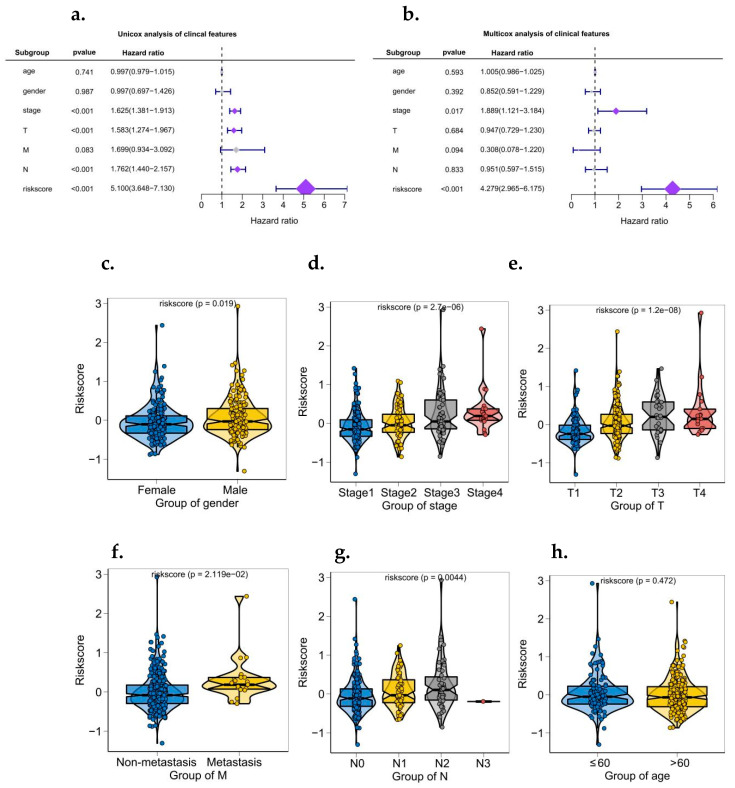
Independent prognostic analysis of a risk scoring model and relationship between risk score and clinical symptoms. (**a**) Univariate analysis of model correlation. (**b**) Multivariate analysis of model correlation. (**c**–**h**) the relationship between risk score and multiple clinical symptoms (gender, stage, T, M, N, age).

**Figure 7 biomedicines-10-02248-f007:**
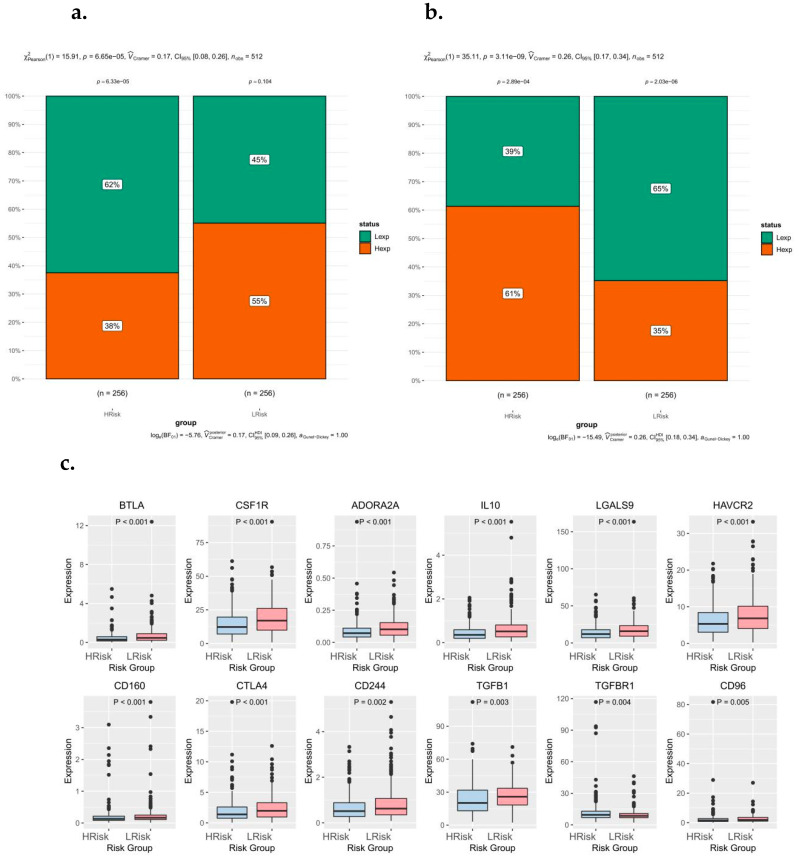
Relationship between risk score subtypes and immunoregulatory factors. (**a**) Differences in immune dysfunction between high and low-risk groups. (**b**) Differences in immune rejection between high and low-risk groups. (**c**) Differences in immune checkpoints and related immunoregulatory factors between high- and low-risk groups.

**Figure 8 biomedicines-10-02248-f008:**
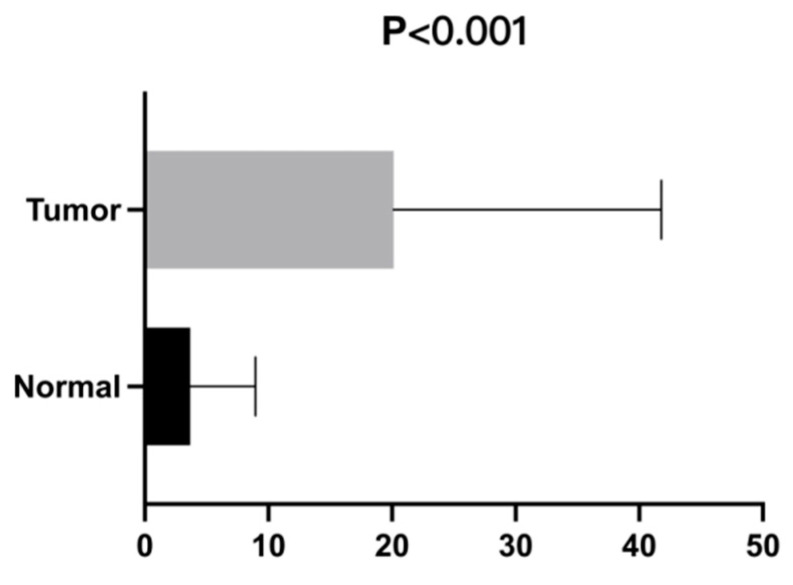
The final QRT-PCR results of TIM3 in lung adenocarcinoma.

**Figure 9 biomedicines-10-02248-f009:**
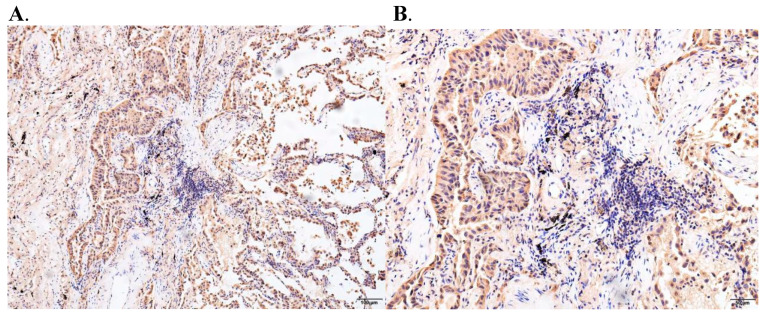
Results of an immunohistochemical assay of TIM3 on lung cancer ((**A**): Stage I; (**B**): Stage II; (**C**): Stage III; (**D**): Stage IV; (**E**): Primary tumor; (**F**): Tumor metastasis).

**Figure 10 biomedicines-10-02248-f010:**
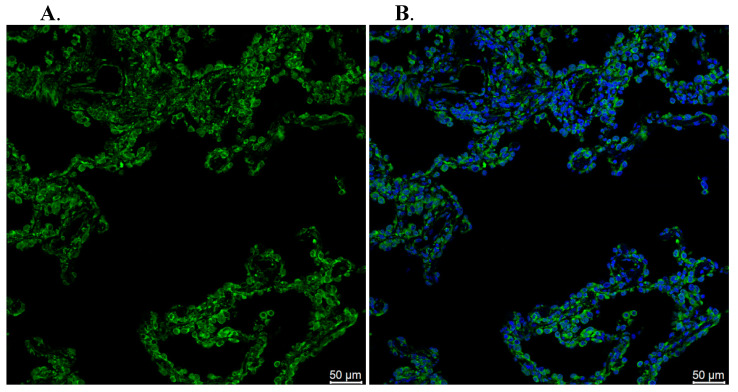
Results of an immunofluorescence assay of TIM3 on lung cancer ((**A**): Stage I; (**B**): Stage II; (**C**): Stage III; (**D**): Stage IV; (**E**): Primary tumor; (**F**): Tumor metastasis).

## Data Availability

Not applicable.

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
