# Peer review of "Immunomodulatory Factor TIM3 of Cytolytic Active Genes Affected the Survival and Prognosis of Lung Adenocarcinoma Patients by Multi-Omics Analysis"

_biomedicines, 2022, doi:10.3390/biomedicines10092248_

Round 1

Reviewer 1 Report

The authors used multi-omics analysis to search for cytolytic active genes associated with survival and prognosis in lung adenocarcinoma. As a result, TIM3 was extracted, and it was found that this gene was highly expressed in tumor tissue and was related to calluses.

The results of the omics analysis are once again collated with individual cases, and it can be seen as a well-conducted study.

What about the following points?

In Figure 7C, a large number of genes are significantly different between the low-risk and high-risk groups. Among them, why did you focus on TIM3/HAVCR2? There are other genes with similar significant differences, and the ones shown in the 7C diagram are all genes related to some form of immune modification.

The reason should be clearly written.

Author Response

Reply to review comments

##Reviewer 1

  1. In Figure 7C, a large number of genes are significantly different between the low-risk and high-risk groups. Among them, why did you focus on TIM3/HAVCR2? There are other genes with similar significant differences, and the ones shown in the 7C diagram are all genes related to some form of immune modification.The reason should be clearly written.

Reply 1. Thank you for your advice, and we have revised and supplemented the content of Figure 7 of the manuscript according to your requirements.

In Figure 7, the genes associated with immune modification include BTLA, CSF1R, ADORA2A, IL-10, LGALS9, HAVCR2, CD160, CTLA4, CD244, TGFB1, TGFBR1 and CD96.

Immune checkpoint receptor protein TIM3 is a type I membrane protein, also known as hepatitis A virus cell receptor 2 (HAVCR2), which is a negative regulator of anti-tumor immunity. A recent article published in the journal Nature on tumor immunotherapy: TIM3 Restrains Anti-tumour Immunity by Regulating Inflammasome Activation, Tumor immunotherapy is a hot spot in current and future research, and T-cell immunoglobulin mucin 3 (TIM-3) is another emerging immune checkpoint molecule after PD-1/PD-L1 and CTLA-4. Tim-3 expression on CD8+ T cells in the tumor microenvironment is considered to be a major marker of T cell dysfunction. However, since tim-3 is also expressed on several other types of immune cells, it is likely that immune cells other than T cells are involved in antitumor effects.

Therefore, TIM3 is of priority for further study among the genes related to immune modification in this study.

Line 322-329

Best wish to you.

Reviewer 2 Report

The Authors conducted a complex study to explore how some immunoregulatory factors affect the prognosis of lung adenocarcinoma.

While the study has some merit, the way it is presented is very confusing. The description of the medical and biological background is over-simplicistic, and the results are very hard to follow, with too many details (is a median score in the test set of -0.0354256887557756 justifable with so many digits?). On the other hand, the Discussion does not present well the significance of the study findings.

I suggest the Authors to carefully reconsider their manuscript.

Author Response

Reply to review comments

##Reviewer 2

  1. While the study has some merit, the way it is presented is very confusing. The description of the medical and biological background is over-simplicistic, and the results are very hard to follow, with too many details (is a median score in the test set of -0.0354256887557756 justifable with so many digits?).

Reply 2.1 Thank you for your advice, and we have simplified part of the results in the manuscript and deleted the more complicated data analysis process, so as to make the results of this study more concise.

Line 181-195.

  1. On the other hand, the Discussion does not present well the significance of the study findings.

Reply 2.2 Thank you for your advice, and we have added the findings of this study and the research significance of TIM3 in the discussion section of the manuscript according to your request. Thank you again for your work.

Line 459-468.

Best wish to you.

Round 2

Reviewer 2 Report

the Authors have done a good work in reviewing the paper. I can only suggest some English polishing